# DOLFIN: DIFFUSION LAYOUT TRANSFORMERS WITH-OUT AUTOENCODER

## ABSTRACT

In this paper, we introduce a novel generative model, Diffusion Layout Transformers without Autoencoder (Dolfin), which significantly improves the modeling capability with reduced complexity compared to existing methods. Dolfin employs a Transformer-based diffusion process to model layout generation. In addition to an efficient bi-directional (non-causal joint) sequence representation, we further propose an autoregressive diffusion model (Dolfin-AR) that is especially adept at capturing rich semantic correlations for the neighboring objects, such as alignment, size, and overlap. When evaluated against standard generative layout benchmarks, Dolfin notably improves performance across various metrics (fid, alignment, overlap, MaxIoU and DocSim scores), enhancing transparency and interoperability in the process. Moreover, Dolfin's applications extend beyond layout generation, making it suitable for modeling geometric structures, such as line segments. Our experiments present both qualitative and quantitative results to demonstrate the advantages of Dolfin.

## 1 INTRODUCTION

Modeling highly-structured geometric scenes such as layout (Zhong et al., 2019) is of both scientific and practical significance in fields like design, 3D modeling, and document analysis. Layout data are typically highly-geometrical-structured, demonstrating strong correlations between neighboring objects in alignment, height, and length. Traditional approaches modeling joint objects using e.g. constellation models (Weber et al., 2000; Fergus et al., 2003; Sudderth et al., 2005) demonstrate interesting results but they typically fail in accurately capturing the joint relations of different objects. Generative models for layout structure in the deep learning era Li et al. (2019); Jyothi et al. (2021); Levi et al. (2023); Inoue et al. (2023); Cheng et al. (2023); Chai et al. (2023) have demonstrated significantly improved quality for the synthesis/generation of layouts.

Before the diffusion model and wide use of transformers, generative modeling is primarily for images with CNN-based models (e.g. VAE (Kingma & Welling, 2013), GAN (Goodfellow et al., 2014)). In the early diffusion image model development, U-Net was commonly used as denoising backbones (e.g. Stable Diffusion (Rombach et al., 2022)), which is not directly applicable to geometric structural data like layouts. A series of work (Levi et al., 2023; Inoue et al., 2023; Cheng et al., 2023; Chai et al., 2023) for layout modeling have therefore adopted the Transformer-based diffusion models (DiT (Peebles & Xie, 2022)) to model token-wise elements such as rectangles in layout. However, they (Levi et al., 2023; Inoue et al., 2023; Cheng et al., 2023; Chai et al., 2023) still follow the original design of DiT, mapping the inputs into a continuous latent space.

In this paper, we present **D**iffusion **l**ayout Trans**f**ormers w**i**thout Autoe**n**coder (Dolfin), that operates on the original space (the coordinates of the bounding box corners and the corresponding class label) for the geometric structural data. Dolfin is a new diffusion model with the following contributions:

- By removing the autoencoder layer that is typically included in a diffusion model for layout/image generation, the Transformers in our model operate directly on the input space of the geometric objects/items (e.g. rectangles), resulting in a generative layout model, Dolfin, that outperforms the competing methods in a number of metrics with reduced algorithm complexity.
- In addition to a bi-directional (non-causal joint) representation for Dolfin, we further propose Dolfin-AR, an autoregressive diffusion model especially effective in capturing the rich semantic correlation between objects/items.

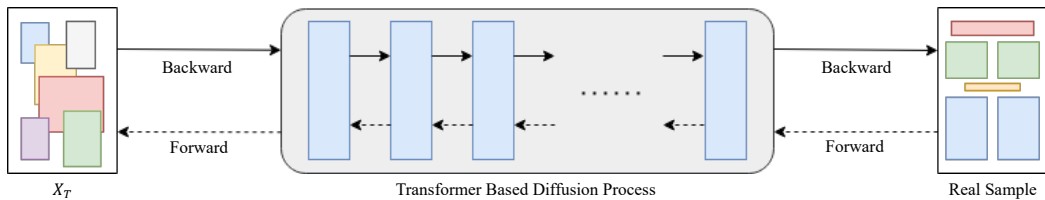

(a) **Dolfin**. The layout tensors are directly fed as input to the transformer-based diffusion block. The model processes the input and generates the desired samples without using an autoregressive decoding process.

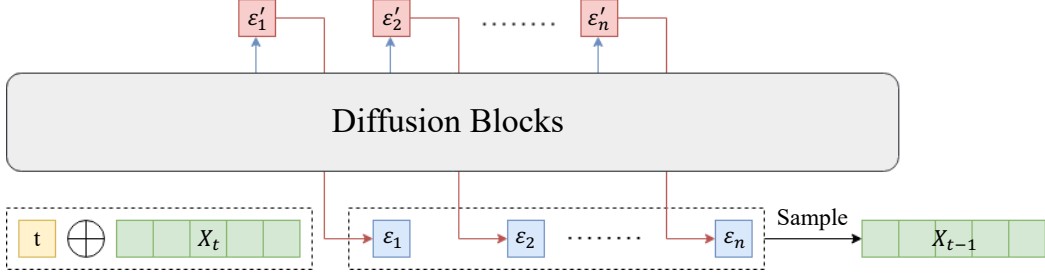

(b) **Dolfin-AR**. The diffusion process starts with the input tensor $x_t$, and it passes through the transformer-based diffusion block. During each autoregressive step $i$, the noise $\epsilon_i$ is sampled, and both $\epsilon_i$ and other inputs are used to sample the next noise $\epsilon_{i+1}$. Finally, the previous sample $x_{t-1}$ is generated based on the sampled noise using DDIM (Song et al., 2022).

Figure 1: The Dolfin model. We directly apply Gaussian noise on the original input space.

- Given the simplicity and generalization capability of the proposed Dolfin model, we experiment on generating geometric structures beyond layout, such as line segments. To the best of our knowledge, this is the first attempt to learn a faithful generative diffusion model for line segment structures that were annotated from natural image scenes.

We make comparisons of our model with other models. In our experiments, our design achieves high performance across numerous metrics.

To further expand the capabilities of our model into line segment generation, an area not yet explored by diffusion models or GAN-based models, our approach demonstrates promising results.

Furthermore, some alternative models, like the SAM algorithm for instance segmentation, directly utilize input tokens in the original space. This aspect makes our approach versatile and suitable for adaptation to various related domains, aligning with the exploding developments of the future.

## 2 BACKGROUND

### 2.1 LAYOUT GENERATION

A layout is characterized by the global height $H$ and width $W$ of the entire scene and a sequence of 5-tuples $\{x_i, y_i, h_i, w_i, c_i\}_{i=1}^{N}$, where each tuple comprises the left-bottom coordinates of a bounding box $(x_i, y_i)$, the height and width $(h_i, w_i)$ of the bounding box, and the corresponding category $c_i$.

In our proposed method, each object in a layout is represented by a $4 \times 4$ tensor. The tensor consists of different entries that encode specific information about the bounding box. The 4 entries on the first row represent $x, y, h, w$ of the bounding box, The entries on the second row denote the height and width of the entire layout. All of these values are normalized to the range $[-1, 1]$. The remaining 8 entries are used to indicate the category of the bounding box. Some of these entries have a value of 1, while others have a value of -1, representing different categories.

The task of layout generation involves generating plausible layouts based on incomplete information through the use of learned models, and it can be classified into two categories: conditional and unconditional. In conditional layout generation, a portion of the layout information is provided, such as the category, shape (height and width), location (coordinates x and y), and other relevant

information. Inspired by the approach used in BLT (Kong et al., 2021), conditionality can be achieved by selectively unmasking specific parts of the tensors, which allows us to generate layouts based on the provided conditions. On the other hand, unconditional layout generation involves generating the final layout entirely from scratch, without any pre-existing conditions or constraints imposed on the layout.

## 2.2 Diffusion Model Used in Layout Generation

A diffusion model learns a prior distribution $p_\theta(x)$ of the target distribution $x$ with parameters $\theta$. It has already been employed in modeling various representations, including 2D images (Ho et al., 2020; Dhariwal & Nichol, 2021; Nichol et al., 2021), videos (Ho et al., 2022b;a), and 3D radiance fields (Shue et al., 2022; Chen et al., 2023). The model consists of a forward diffusion process and a backward denoising process. The forward process is modeled by a Markov chain which gradually adds Gaussian noise with scheduled variances to the input data. The noisy data at time step $t$ has the closed form representation as the following:

$$x_t = \sqrt{\bar{\alpha}_t}x_0 + \sqrt{1 - \bar{\alpha}_t} \cdot \epsilon_t, \quad \epsilon_t \sim \mathcal{N}(0, \mathbf{I}) \tag{1}$$

where $\bar{\alpha}_t$ is a constant determined by $t$. Based on Equation 1, we can directly sample $x_t$ from $x_0$ during training.

In the backward process, a denoising network is learned for predicting and removing the noise of noisy data $x_t$ at each time step. The network is optimized by an L2 denoising loss:

$$L_{\text{diff}} = \|\epsilon_t - \epsilon_\theta(x_t)\|_2^2 \tag{2}$$

where $\epsilon_\theta(x_t)$ is the noise predicted by our model with model parameter $\theta$ as well as input $x_t$ and $t$.

## 3 Method

We propose Dolfin, a diffusion layout transformer for generative layout and structure modeling. Figure 1 provides an overview of Dolfin.

## 3.1 Diffusion Layout Transformers

As mentioned in Section 2.1, our input data for layout generation comprises both continuous components (bounding boxes) and discrete components (categories). In contrast to previous methods that encode the input into a continuous latent space or separate the discrete/continuous components and design a specialized discrete diffusion process for discrete parts, our method directly operates on the input geometric objects without any additional modules or modifications on the diffusion model.

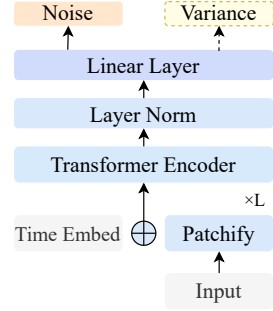

Inspired by the recent work DiT (Peebles & Xie, 2022), we select a transformer based diffusion model as the base architecture of Dolfin, as transformer-based models are particularly good at modeling sequential data. As described before, we directly process the data in the discrete input space by adding Gaussian noise to the input tensor. The transformer network predicts the noise at each step of the diffusion process.

Figure 2: The transformer structure consists of L transformer layers as well as a layer norm and a linear layer.

Starting from the original DiT, we make several modifications to the transformer model. Our model only embeds the timestamp $t$ into the input, as opposed to embedding both the timestamp $t$ and a global category $c$. Figure 2 illustrates the structure.

Our model consists of two versions: the non-autoregressive version and the autoregressive version. The key difference between the two is the training approach used for the transformer encoder. In the non-autoregressive version, the transformer encoder is trained in a non-autoregressive way. Meanwhile, in the autoregressive version, the transformer encoder is trained in an autoregressive way.

## 3.2 Non-Autoregressive Model

In this version of our method, the transformer operates in a non-autoregressive manner, processing all tokens simultaneously rather than sequentially. Figure 1a provides an overview of our method.

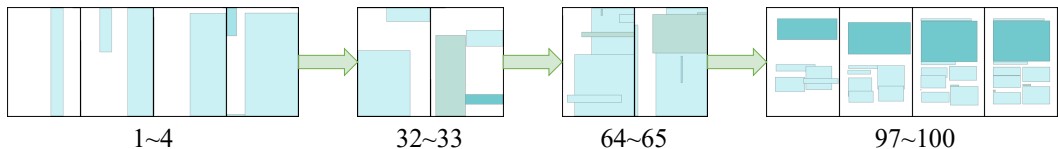

| 1~4 | 32~33 | 64~65 | 97~100 |

Figure 3: The figure shows the intermediate results of the diffusion process, with numerical labels indicating the number of diffusion steps completed by the corresponding subfigures. Each subfigure represents a diffusion fragment, arranged from left to right.

### 3.2.1 TRAINING

Given an initial layout represented by a vector $x_{start}$ and a diffusion timestep $t$, we randomly sample a Gaussian noise $\epsilon_t$ and apply **Equation (1)** to add noise to $x_{start}$, resulting in $x_t$. Next, we input $x_t$ and $t$ into a transformer model to predict the noise $\epsilon_\theta(x_t)$ and the variance of the step (only required for DDPM (Ho et al., 2020), not required for DDIM in the DiT (Peebles & Xie, 2022) framework) $\hat{\sigma}_t$. The mean squared error (MSE) loss is then calculated between $\epsilon_0$ and $\epsilon_\theta(x_t)$, while the Kullback-Leibler (KL) divergence loss is calculated (if we use DDPM) between the actual and predicted mean and variance $\mu_t, \sigma_t$ and $\hat{\mu}_t, \hat{\sigma}_t$, where $\hat{\mu}_t$ can be sampled by $\hat{\sigma}_t$ and $\epsilon_\theta(x_t)$ in DDPM.

$$\mathcal{L}_{MSE} = \|\epsilon_t - \epsilon_\theta(x_t)\|_2^2 \tag{3}$$

$$\mathcal{L}_{KL} = \sum_t D_{\mathrm{KL}}\left(q\left(\mathbf{x}_{t-1} \mid \mathbf{x}_t, \mathbf{x}_0\right) \| p_\theta\left(\mathbf{x}_{t-1} \mid \mathbf{x}_t\right)\right) = \sum_t D_{KL}(\mu_t, \sigma_t \| \hat{\mu}_t, \hat{\sigma}_t) \tag{4}$$

### 3.2.2 SAMPLING

Let $T$ be the number of diffusion steps. At each time step $t$ from $T-1$ to 0, we first input $x_{t+1}$ and timestamp $t+1$ into the pre-trained transformer to obtain the predicted noise $\epsilon_\theta(x_{t+1})$. We then use DDPM to compute $x_t$ from $x_{t+1}$ and $\epsilon_\theta(x_{t+1})$. This process is repeated until the final sample $x_0$ is obtained.

### 3.3 AUTOREGRESSIVE MODEL

In contrast to the non-autoregressive model that samples $\hat{\epsilon}$ using a single step, our proposed method adopts a recursive sampling strategy repeated for $N$ times, where $N$ corresponds to the number of tokens. This approach allows for more comprehensive sampling and captures the dependencies among tokens. The training and sampling procedures are outlined in the following pseudocode, and Figure 1b offers a visual representation of this approach.

---

**Algorithm 1:** Autoregressive Training

**Input:** input $x_0$, timestamp $t$
Sample $x_t$ from $x_0$ and $t$;
**for** $t = 0$ *to* $N-1$ **do**
    $n\_in = concat\left(x_t, noise[0 \cdots t-1]\right)$;
    $\epsilon_\theta(x_t) = Model(n\_in, t)$;
    $Loss = MSE(\epsilon_\theta(x_t), noise[t])$;
    Update the parameters
      $f \leftarrow f - \eta\nabla_f Loss$;
**end**

---

**Algorithm 2:** Autoregressive Sampling

**Input:** input $x_t$, timestamp $t$
$x = x_t$;
**for** $t = 0$ *to* $N-1$ **do**
    $noise[t] = Model(x, t)$;
    $x = concat(x, noise[t])$;
**end**
$\epsilon_\theta(x_t) = noise[0 \cdots N-1]$;
Sample $x_{t-1}$ by $x_t$, $t$, $\epsilon_\theta(x_t)$ using
  DDIM (Song et al., 2022);

---

### 3.4 COMPARISON OF OUR METHOD WITH OTHER APPROACHES

Our model has several advantages over other methods in the field. One of the main benefits is that it employs a relatively simple model, which makes it computationally efficient. In particular, unlike the works PLay Cheng et al. (2023) and DLT (Levi et al., 2023), in which the inputs are encoded to a latent space, diffusion is directly applied to the original space. What's more, this approach eliminates the need to separate each object in the layout into several different tokens like the method provided in LayoutDM (Inoue et al., 2023), which in turn requires less computation.

Our method offers a notable advantage in terms of the transparency of the diffusion sampling process. By directly applying the diffusion model to the original space, we can observe the step-by-step transformation of a randomly sampled Gaussian noise into a meaningful and coherent layout.

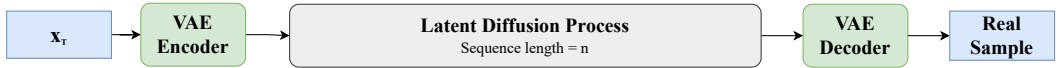

(a) In the redrawing of PLay (Cheng et al., 2023), the input layout is encoded using a pretrained Variational Autoencoder (VAE) as a first-stage model to map it into a latent space. The diffusion process is then applied, and the final result is generated by decoding the latent representation using a VAE decoder.

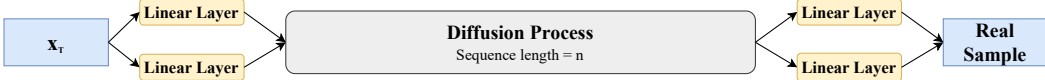

(b) In the redrawing of DLT (Levi et al., 2023), the attributes of a layout are divided into discrete and continuous parts, and they are processed separately through different linear layers before being concatenated and fed into the diffusion process. The output of the diffusion process is then passed through separate linear layers to obtain the final sample.

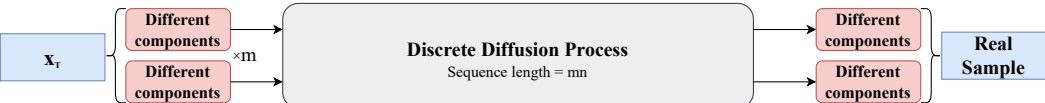

(c) In the redrawing of LayoutDM (Inoue et al., 2023), a layout is divided into multiple discrete components, resulting in a token sequence with a length of $mn$, where $m$ represents the number of components and $n$ is the number of objects in a layout. This differs from other methods, which typically have a token sequence length equal to the number of items in a layout.

Figure 4: The comparison between different layout generating models

This transparency allows for a better understanding of the generative process and facilitates the interpretation of the generated layouts.

To visually depict this diffusion sampling process, we present Figure 3, which showcases the gradual refinement of the layout from its initial random state to its final coherent form. The figure serves as an illustration of the step-by-step evolution of the layout during the diffusion process, highlighting the transition from noise to a structured and visually pleasing arrangement. This visual representation enhances our understanding of the generative process and provides insights into the underlying mechanisms driving the layout generation.

### 3.5 BEYOND LAYOUT GENERATION

Dolfin can be extended beyond simple application of layout generation. As the transformer in Dolfin directly operates on the input geometric objects, it can manage different types of geometric structures.

We present one application on line segments generation. Dolfin is able to learn distributions of line segments from the images in the training set. Similar to layout generation, we take two endpoint coordinates of each line as one input token to the transformer, directly adding noise to the tokens and performing denoising. By sampling from the learned distribution, the model generates novel line segments that are natural and plausible.

### 4 RELATED WORK

**Layout Generation**. The task of layout generation involves using generative models to create layouts (vectorized data) for various purposes such as houses, rooms, posters, documents, and user interfaces (Nauata et al., 2020; 2021; Deka et al., 2017b; Fu et al., 2022; Guo et al., 2021; Kikuchi et al., 2021; Yamaguchi, 2021; Yang et al., 2016; Zheng et al., 2019; Singh et al., 2022; Shabani et al., 2022). LayoutGAN (Li et al., 2019) and LayoutVAE (Jyothi et al., 2021) are two representative methods. In LayoutGAN, a generator is trained to produce layouts using a differentiable wireframe rendering layer, while a discriminator is trained to assess the alignment. This approach allows for the generation of realistic layouts. LayoutVAE employs a variational autoencoder (VAE) (Kingma & Welling, 2022) to generate layouts. The VAE utilizes a Long Short-Term Memory (LSTM) network (Hochreiter & Schmidhuber, 1997) to extract relevant information for generating layouts. Recent works, such as BLT (Kong et al., 2021), introduce a bidirectional transformer for layout generation, which further enhances the model capacity of extracting relationships between different layout objects. The work LT (Gupta et al., 2021) introduces an autoregressive model without diffusion, tokens in a sequence are sampled one by one, allowing each token to incorporate information from the

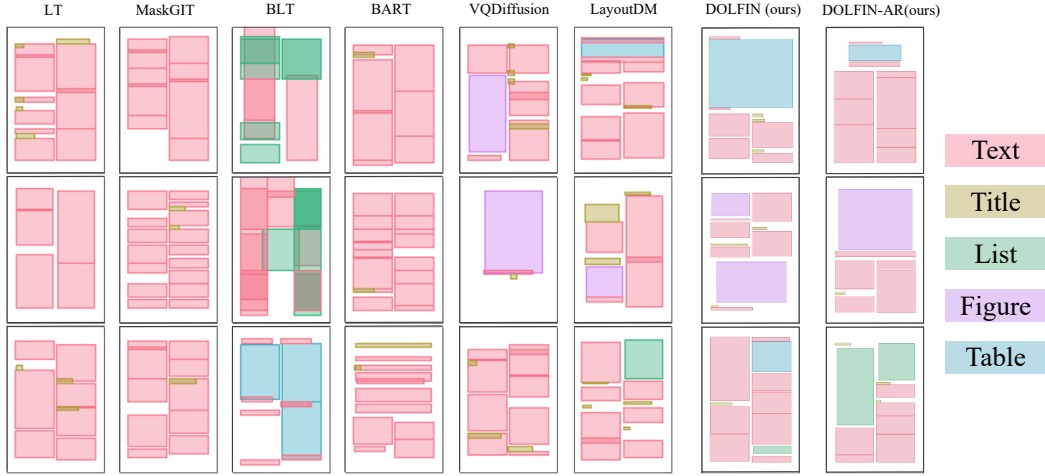

Figure 5: Unconditional generation on the PublayNet dataset. The first 6 columns are from paper LayoutDM (Inoue et al., 2023), each including 3 visual results of unconditional generation by LT (Gupta et al., 2021), MaskGIT (Chang et al., 2022), BLT (Kong et al., 2021), BART (Lewis et al., 2019), VQDiffusion (Gu et al., 2022) and LayoutDM (Inoue et al., 2023) respectively. The last 2 columns contain the visual results of our Dolfin and Dolfin-AR. methods

previously sampled tokens. DLT (Levi et al., 2023), LayoutDM (Inoue et al., 2023), and PLay (Cheng et al., 2023), introduce transformer-based (Vaswani et al., 2017) diffusion processes, resulting in improved performance and layout generation capabilities.

In real-world applications, the generated layouts are often required to meet customized constraints like categories or partial layouts, which yields the task of conditional layout generation. Kikuchi et al. (2021); Arroyo et al. (2021); Kong et al. (2021); Inoue et al. (2023); Levi et al. (2023); Chai et al. (2023) Particular conditions include categories, bounding box numbers, fixed bounding box positions and sizes, etc. Conditions are added by using some encoders to encode the constraints or simply by applying simple masks.

**Layout Generation Using Diffusion Models**. With the rapid development of diffusion models (Ho et al., 2020; Song et al., 2022), the diffusion model is being applied in layout generating tasks in recent days. Currently, researchers employ three main approaches to address this problem. In PLay (Cheng et al., 2023) (Figure 4a), an encoder is used to map the layout into a continuous latent space. Diffusion models are then applied to this latent space instead of the original space. In DLT (Levi et al., 2023) (Figure 4b), separate linear layers are incorporated for the continuous and discrete parts of the input tensor, both before and after the transformer block. When adding noise during the diffusion process, typical Gaussian noise is applied to the continuous parts, while for the discrete parts, a discrete diffusion process is defined to introduce noise. In LayoutDM (Inoue et al., 2023) (Figure 4c), layouts are considered as compositions of distinct discrete elements, such as category, size, and position. To enable diffusion modeling in this context, a discrete version of the diffusion process is devised. What's more, the sequences that enter the transformer have length far larger than the number of items in the layouts, which is computational consuming.

In our method, as shown in Figure 1, we simply view each element as a whole and directly add Gaussian noise on the original vector space without projecting it to a latent space.

## 5 EXPERIMENTS

### 5.1 IMPLEMENTATION DETAILS

#### 5.1.1 DATASETS

Our model is trained using two widely used layout datasets on document and user interface, PublayNet (Zhong et al., 2019) and RICO (Deka et al., 2017a). The PublayNet is a large dataset of document layouts which includes around 330,000 samples extracted from published scientific papers. The dataset is categorized into five classes: text, title, figure, list, and table. It is divided into training, validation, and testing sets as described in (Zhong et al., 2019). The RICO dataset consists of about 70,000 layouts of user interface designs, classified into 25 categories. We adopt the 85%-5%-10% split for training, validation, and testing as in (Inoue et al., 2023; Levi et al., 2023).

| Method | PublayNet Align. ↓ | PublayNet Overlap ↓ | PublayNet FID ↓ | PublayNet MaxIoU ↑ | RICO Align. ↓ | RICO FID ↓ | RICO MaxIoU ↑ |
|---|---|---|---|---|---|---|---|
| Real Data | 0.021 | 4.2 | - | - | 0.109 | - | - |
| PLay (Cheng et al., 2023) | - | - | 13.71 | - | - | **13.00** | - |
| VTN (Arroyo et al., 2021) | - | 2.6 | 19.80 | - | - | 18.80 | - |
| LayoutDM (Inoue et al., 2023) | 0.195 | - | - | - | 0.162 | - | - |
| LT-fixed (Gupta et al., 2021) | 0.084 | - | - | - | 0.133 | - | - |
| LT (Gupta et al., 2021) | 0.127 | **2.4** | - | - | 0.068 | - | - |
| MaskGIT (Chang et al., 2022) | 0.101 | - | - | - | **0.015** | - | - |
| BLT (Kong et al., 2021) | 0.153 | 2.7 | - | - | 1.030 | - | - |
| BART (Lewis et al., 2019) | 0.116 | - | - | - | 0.090 | - | - |
| VQDiffusion (Gu et al., 2022) | 0.193 | - | - | - | 0.178 | - | - |
| DLT (Levi et al., 2023) | 0.110 | 2.6 | - | - | 0.210 | - | - |
| LayoutGAN-W (Li et al., 2019) | - | - | - | 0.21 | - | - | 0.24 |
| LayoutGAN-R (Li et al., 2019) | - | - | - | 0.24 | - | - | 0.30 |
| NDN-none (Gu et al., 2022) | - | - | - | 0.31 | - | - | 0.35 |
| LayoutGAN++ (Kikuchi et al., 2021) | - | - | - | **0.36** | - | - | 0.36 |
| Dolfin (**ours**) | 0.074 | 5.0 | **9.12** | 0.35 | 0.094 | 15.38 | **0.42** |
| Dolfin-AR (**ours**) | **0.042** | **2.4** | 16.88 | **0.36** | 0.148 | 26.44 | 0.17 |

Table 1: Quantitative results of unconditional generation on PublayNet and RICO. The best result are highlighted in **bold**. Many numbers are not included in the corresponding or related papers so that we do not show them here. For FID score, it is unclear about some details (e.g. the number of samples to use when comparing with the testing set, which is sensitive to the FID score) in many of the works, making it difficult to compare the numerical results with the original paper on the FID metric. On PublayNet dataset, which is highly-geometrical-structural, our method provides general superior results over the metrics.

### 5.1.2 TRAINING DETAILS

The transformer encoder in Dolfin consists of 4 attention layers and 8 attention heads with a hidden size of 512. This parameter setting makes our model size similar to other baselines, ensuring a fair comparison. The number of diffusion steps is set to 100. The AdamW optimizer is employed with a learning rate of 1e-4. The batch size is set to 10000 for the training of non-autogressive model and 6000 for the training of autogressive model.

Dolfin is trained on 8 NVIDIA RTX A5000 GPUs. It takes approximately 48 hours to train the non-autoregressive model and 96 hours to train the autoregressive model.

### 5.1.3 METRICS

We employ five different metrics to evaluate the quality of the generated layouts on the PublayNet dataset: Fréchet Inception Distance (FID) score (Heusel et al., 2018), overlap score, alignment score, MaxIoU score and DocSim score. For the RICO dataset, we do not calculate the overlap score as the data samples in RICO typically contain large regions of overlap.

To compute the FID score, we first render the generated layouts into images. Subsequently, we utilize a widely accepted Inception model (Szegedy et al., 2017) to measure the similarity between the generated images and the real images from the dataset. The computation process follows the established methodology presented in the work PLay (Cheng et al., 2023), ensuring a fair comparison with their results.

For overlap score and alignment score, we adopt similar evaluation protocols as those employed in previous works. Specifically, we utilize data from the LayoutGAN++ (Kikuchi et al., 2021), DLT (Levi et al., 2023) and LayoutDM (Inoue et al., 2023) papers to compute the scores. These metrics provide objective measures of the quality and accuracy of the generated layouts.

Notably, the datasets exhibit different characteristics in terms of alignment and overlap. While the PublayNet dataset demonstrates excellent alignment, it has relatively little overlap between objects. In contrast, the RICO dataset contains numerous instances of overlap between objects. As a result, we do not use overlap as a metric on the dataset RICO.

### 5.2 EXPERIMENTAL RESULTS

We compare the results of Dolfin and Dolfin-AR with several other methods list in the tables. We present the comparisons of unconditional generation performances in Table 1, PublayNet on the left and RICO on the right. Some baselines do not provide the results on all the metrics, so we only report the available values included in their original papers. The results shows that the proposed Dolfin model, including both the autoregressive and non-autoregressive versions, consistently outperforms other SO on various tasks, except for the results of unconditional generation on the RICO dataset.

Table 2: MaxIoU scores for models conditioning on category / category+size on PublayNet and RICO. Our model achieves superior performance and outperform the competing methods in most cases. This validates the efficacy of incorporating Transformer based architecture into Dolfin, which enables directly operating on the original space of rectangles (geometric structures).

| Method | PublayNet Cate. ↓ | PublayNet Cate.+Size ↓ | Rico Cate. ↓ | Rico Cate.+Size ↓ |
|---|---|---|---|---|
| LayoutVAE (Jyothi et al., 2021) | 0.316 | 0.315 | 0.249 | 0.283 |
| NDN-none (Lee et al., 2020) | 0.162 | 0.222 | 0.158 | 0.219 |
| LayoutGAN++ (Kikuchi et al., 2021) | 0.263 | 0.342 | 0.267 | 0.348 |
| LT (Gupta et al., 2021) | 0.272 | 0.320 | 0.223 | 0.323 |
| MaskGIT (Chang et al., 2022) | 0.319 | 0.380 | 0.262 | 0.320 |
| BLT (Kong et al., 2021) | 0.215 | **0.387** | 0.202 | 0.340 |
| BART (Lewis et al., 2019) | 0.320 | 0.375 | 0.253 | 0.334 |
| VQDiffusion (Gu et al., 2022) | 0.319 | 0.374 | 0.252 | 0.331 |
| LayoutDM (Inoue et al., 2023) | 0.310 | 0.381 | 0.277 | 0.392 |
| LayoutDM(1) (Chai et al., 2023) | **0.440** | - | 0.490 | - |
| Dolfin (ours) | 0.336 | 0.339 | **0.529** | **0.550** |

Table 3: Quantitative results of line segment generation. Dolfin performs better than the competing methods.

| Method | FID ↓ | Difference ↓ |
|---|---|---|
| LT (Gupta et al., 2021) | 317.4 | 30.86 |
| LayoutGAN++ (Kikuchi et al., 2021) | 400.0 | 51.90 |
| Dolfin (Ours) | **92.1** | **16.04** |

Table 4: Alignment scores for models conditioning on category / category+size on PublayNet and RICO. Our method performs well when comparing with other baselines.

| Method | PublayNet Cate. ↓ | PublayNet Cate.+Size ↓ | Rico Cate. ↓ | Rico Cate.+Size ↓ |
|---|---|---|---|---|
| Real Data | 0.02 | 0.02 | 0.11 | 0.11 |
| VTN (Arroyo et al., 2021) | 0.29 | 0.09 | 0.43 | 0.44 |
| BLT (Kong et al., 2021) | **0.10** | 0.09 | **0.12** | 0.30 |
| LT (Gupta et al., 2021) | 0.41 | 0.14 | 0.58 | 0.41 |
| DLT (Levi et al., 2023) | 0.11 | 0.09 | 0.18 | 0.28 |
| LayoutDM(1) (Chai et al., 2023) | 0.15 | - | 0.36 | - |
| Dolfin (**ours**) | 0.41 | **0.08** | 0.17 | **0.14** |

Table 5: DocSim scores for models conditioning on category / category+size on PublayNet and RICO. We observe a significant boost of performance by our method when compared with existing approaches.

| Method | PublayNet Cate. ↓ | PublayNet Cate.+Size ↓ | Rico Cate. ↓ | Rico Cate.+Size ↓ |
|---|---|---|---|---|
| LayoutVAE (Jyothi et al., 2021) | 0.07 | 0.09 | 0.13 | 0.19 |
| NDN-none (Lee et al., 2020) | 0.06 | 0.09 | 0.15 | 0.21 |
| LT (Gupta et al., 2021) | 0.11 | - | 0.20 | - |
| VTN (Arroyo et al., 2021) | 0.10 | - | 0.20 | - |
| BLT (Kong et al., 2021) | 0.11 | 0.18 | 0.21 | 0.30 |
| Dolfin (ours) | **0.42** | **0.42** | **0.57** | **0.59** |

The reason for this slight disadvantage may be due to our method being more suitable on layout data with well-aligned and non-overlapping bounding boxes, while the data in RICO contains large regions of overlap.

For conditional generation, we utilize the MaxIoU score, alignment score and DocSim score as the metrics. The corresponding results are shown in Table 2, Table 4 and Table 5 respectively. We test on sampling with conditions include category and category+size (height and width of the bounding boxes) on both PublayNet and RICO datasets. The data shows that even on dataset with large regions of overlap, our method can still achieve nice performance.

We also illustrate qualitative generations (unconditional) results in Figure 5, which shows that our methods can efficiently generate satisfying layouts. For more results of conditional and unconditional generation, please refer to the supplementary material.

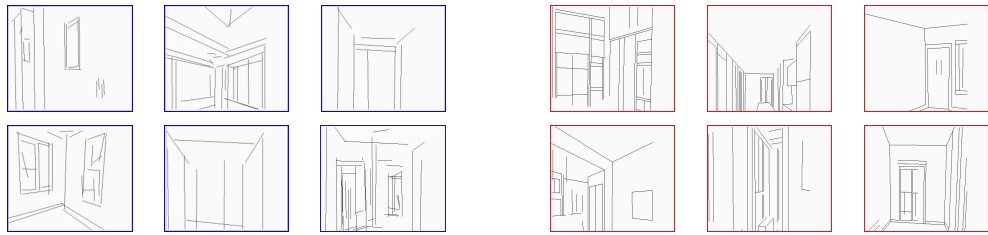

Figure 6: Examples of line segments generation. Images with blue frames are generated results and those with red frames are samples from training set.

### 5.3 LINE SEGMENTS GENERATION

We train our non-autoregressive Dolfin model on the ShanghaiTech Wireframe dataset (Huang et al., 2018), which consists of 5000 training images. We extract the line segments from each image and use them directly as the input of Dolfin for training.

We present line segments generation results in Figure 6. We also show the output of ControlNet (Zhang & Agrawala, 2023) based on the line segments for better illustration in the supplementary material. For quantitative results and comparisons, we have diligently searched for relevant works in the

Table 6: Unconditional Generation On PublayNet Dataset (Using MLP).

| Method | Align. ↓ | FID ↓ |
|---|---|---|
| Dolfin (MLP Autoencoder) | 0.129 | 15.50 |
| Dolfin (Regular) | **0.074** | **9.12** |

Table 7: Unconditional generation on PublayNet dataset using different transformer structure

| Transformer | $4 \times 384$ | $4 \times 512$ | $8 \times 512$ |
|---|---|---|---|
| **FID Score** | 9.49 | 9.12 | 9.45 |
| **Alignment** | 0.062 | 0.074 | 0.057 |

same domain, but regrettably, we could not identify any. To make comparisons, we reimplement other models to handle the task as our baseline models, specifically LT (Gupta et al., 2021) and LayoutGAN++ (Kikuchi et al., 2021). Our proposed model demonstrates superior line segment generation results compared to these baselines. For a comprehensive evaluation, we utilize the FID score (Heusel et al., 2018) and an additional metric (Difference) as detailed in supplementary material. The numerical results are shown in Table 3

## 6 ABLATION STUDY

To show the effectiveness of our method Dolfin as well as to reveal the impacts of some parameters, we conduct several ablation studies. (1) Compare the results of encoding the input to latent space and directly operating on input space. (2) Compare the results of using different transformer architectures. (3) Measure the time cost of sampling a layout using our model with different number of tokens.

### 6.1 DIRECT OPERATE ON THE INPUT SPACE

We extend our model by incorporating a two-stage architecture, where we introduce a Multilayer Perceptron (MLP) as an autoencoder. This MLP is trained to map the input tensor into a latent space representation. By leveraging this two-stage model, our approach operates in the latent space, similar to the methodology proposed in the PLay (Cheng et al., 2023) framework. Instead of directly sending the tensor into the transformer block, we initially encode it using the MLP, obtaining a latent representation, which is subsequently fed into the transformer block for further processing.

We evaluate the performance of this extended model through quantitative analysis, as presented in Table 6. The results indicate that the model with the MLP performs significantly worse compared to the model without it. This suggests that the introduction of the MLP autoencoder does not yield improvements in terms of the evaluated metrics.

### 6.2 ADJUSTING TRANSFORMER ARCHITECTURES

We conducted experiments with different transformer structures, including $4 \times 384$ and $8 \times 512$, in addition to the original $4 \times 512$ transformer block. The purpose was to explore the impact of varying the number of layers and hidden dimensions on the performance of our method. By evaluating the unconditional generation FID and alignment score on PublayNet dataset, we observed that increasing the number of transformer parameters does not necessarily lead to improved performance. The results are presented in Table 7.

Table 8: Time cost for generating a single layout with our two models on the two datasets.

| Model | PublayNet | RICO |
|---|---|---|
| Dolfin | 0.021 | 0.303 |
| Dolfin-AR | 0.029 | 0.375 |

### 6.3 TIME COST OF OUR MODEL

We record the time cost per sample on a single A5000 GPU of Dolfin and Dolfin-AR on datasets PublayNet and RICO. Each PublayNet sample consists of 16 tokens, while RICO sample has 25. The results are shown in Table 8. Although AR model uses longer inference time, it is still acceptable.

## 7 CONCLUSION

In this paper, we propose Dolfin, a novel diffusion layout transformer model for layout generation. Dolfin directly operates on the discrete input space of geometric objects, which reduces the complexity of the model and improves efficiency. We also present an autoregressive diffusion model which helps capture semantic correlations and interactions between input transformer tokens. Our experiments demonstrate the effectiveness of Dolfin that improves current state-of-the-arts across multiple metrics. In addition, Dolfin can be extended beyond layout generation, e.g. to modeling other geometric structures such as line segments.

**Limitation** The advantage of Dolfin in modeling layout that is less-geometrical-structured (e.g. the RICO data) is not as obvious as that in modeling highly-geometrical-structured (e.g. the PublayNet).

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

APPENDIX (SUPPLEMENTARY MATERIAL)

## A  VISUAL COMPARISON WITH OTHER METHODS

We provide qualitative comparisons in Figure 7 and 5. We compare the proposed model Dolfin with other state-of-the-art layout generation methods including DLT (Levi et al., 2023), BLT (Kong et al., 2021), VTN (Arroyo et al., 2021), LT (Gupta et al., 2021), MaskGIT (Chang et al., 2022), BART (Lewis et al., 2019), VQDiffusion (Gu et al., 2022) and LayoutDM (Inoue et al., 2023). We use the same image samples as in papers DLT (Levi et al., 2023) and LayoutDM (Inoue et al., 2023).

### A.1  CONDITIONAL GENERATION ON PUBLAYNET DATASET

We compare the conditional generation results of Dolfin with DLT (Levi et al., 2023), BLT (Kong et al., 2021), VTN (Arroyo et al., 2021) and LT (Gupta et al., 2021). The conditions are set to category and category + size (height and width). Results are presented in Figure 7.

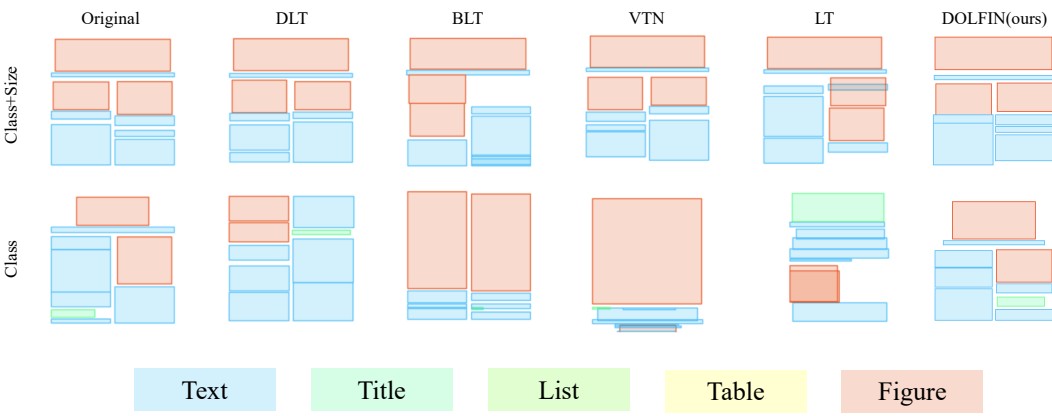

Figure 7: The first 5 columns are copied from paper DLT Levi et al. (2023), including visual results of category conditional generation and category+size conditional generation of methods DLT Levi et al. (2023), BLT Kong et al. (2021), VTN Arroyo et al. (2021) and LT Gupta et al. (2021). The first column contains the images from PublayNet dataset. The last column contains the results by our Dolfin method.

### A.2  UNCONDITIONAL GENERATION ON PUBLAYNET DATASET

We compare the unconditional generation results of Dolfin with LT Gupta et al. (2021), MaskGIT Chang et al. (2022), BLT Kong et al. (2021), BART Lewis et al. (2019), VQDiffusion Gu et al. (2022) and LayoutDM Inoue et al. (2023). Results are shown in Figure 8. Each model is provided with 5 different generation results for comparing both generation quality and diversity.

## B  DOLFIN UNCONDITIONAL GENERATION ON OTHER DATASETS

### B.1  COCO DATASET

COCO dataset (Lin et al., 2015) is a large-scale object detection, segmentation, and captioning dataset with more than 330K images. We use the bounding boxes in the dataset as our data. We adopt the official datasplit of the dataset. We present the numerical results for 2 metrics(alignment, DocSim) on the dataset in Table9, and our model achieves better performance than the competing methods. The visual results are shown in Figure9.

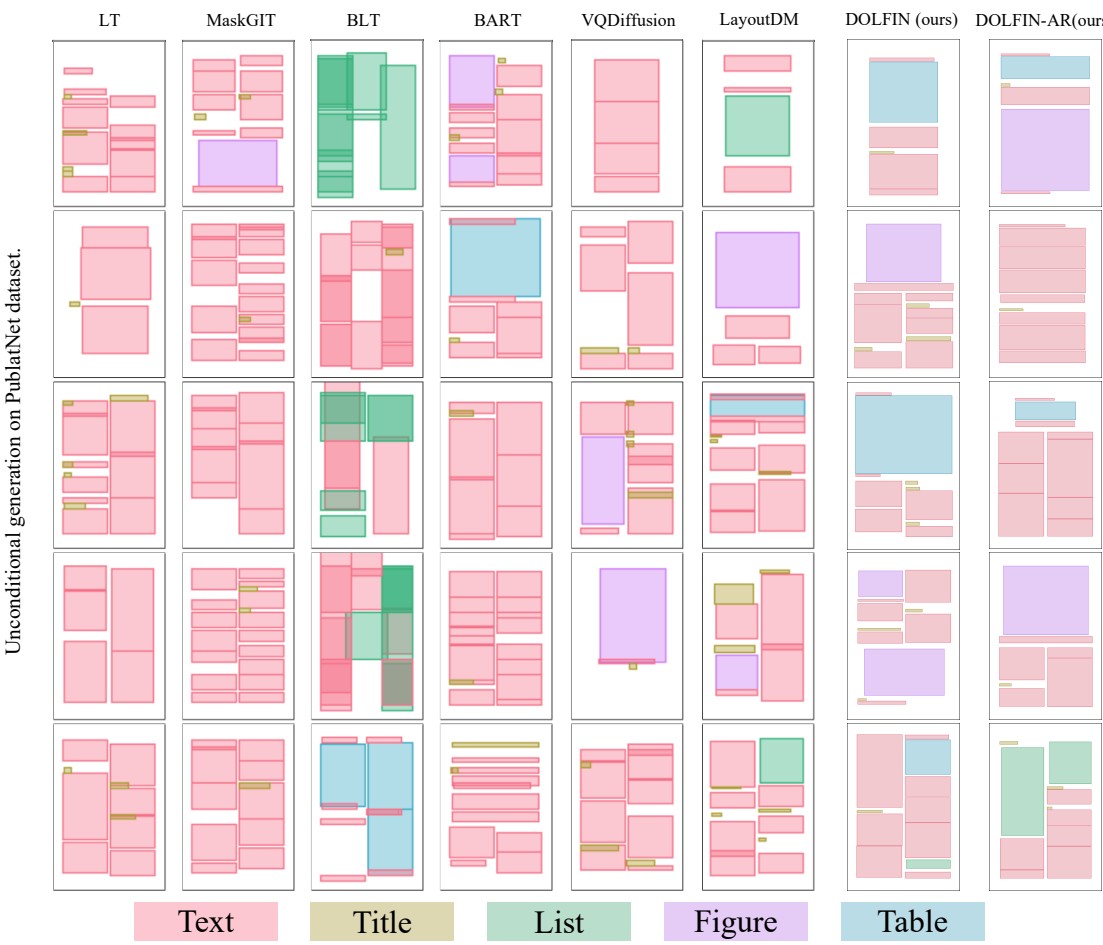

Figure 8: The first 6 columns are copied from paper LayoutDM Inoue et al. (2023), each including 5 visual results of unconditional generation by LT Gupta et al. (2021), MaskGIT Chang et al. (2022), BLT Kong et al. (2021), BART Lewis et al. (2019), VQDiffusion Gu et al. (2022) and LayoutDM Inoue et al. (2023) respectively. The last 2 columns contain the visual results of our Dolfin and Dolfin-AR. methods

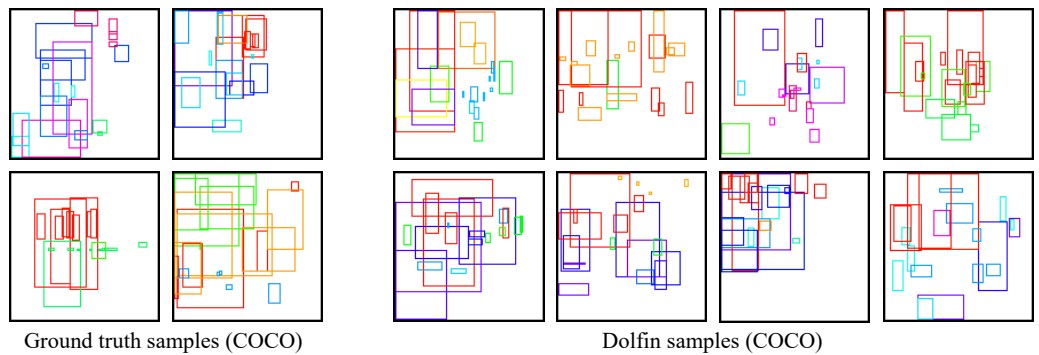

Figure 9: The left 4 figures are sampled from the COCO dataset while the right 4 figures are generated results. We can see that our samples satisfy both quality and diversity.

Table 9: COCO dataset quantitative result.

| Method | Align. ↓ | DocSim ↑ |
|---|---|---|
| LayoutVAE (Jyothi et al., 2021) | 0.246 | - |
| LayoutTrans. (Gupta et al., 2021) | 0.334 | - |
| VTN (Arroyo et al., 2021) | 0.330 | - |
| BLT (Kong et al., 2021) | 0.311 | - |
| Dolfin (Ours) | 0.331 | 0.14 |
| Dolfin-AR (Ours) | **0.215** | 0.15 |

Table 10: Magazine dataset quantitative result.

| Method | MaxIoU ↑ | Align. ↓ | DocSim ↑ |
|---|---|---|---|
| LayoutGAN-W (Li et al., 2019) | 0.12 | 0.74 | - |
| LayoutGAN-R (Li et al., 2019) | 0.16 | 1.90 | - |
| NDN-none (Gu et al., 2022) | 0.22 | 1.05 | - |
| LayoutGAN++ (Kikuchi et al., 2021) | 0.26 | 0.80 | - |
| Dolfin (Ours) | **0.63** | 0.76 | 0.11 |
| Dolfin-AR (Ours) | 0.50 | **0.39** | 0.08 |

## B.2 MAGAZINE DATASET

Magazine dataset (Xinru Zheng & Lau, 2019) is a large and diverse magazine layout dataset with 3,919 images of magazine layouts. We adopt the official datasplit of the dataset. We present the numerical results for 3 metrics(MaxIoU, alignment, DocSim) on the dataset in Table10. Our model achieves superior performance, outperforming the competing methods. The visual results are shown in Figure10.

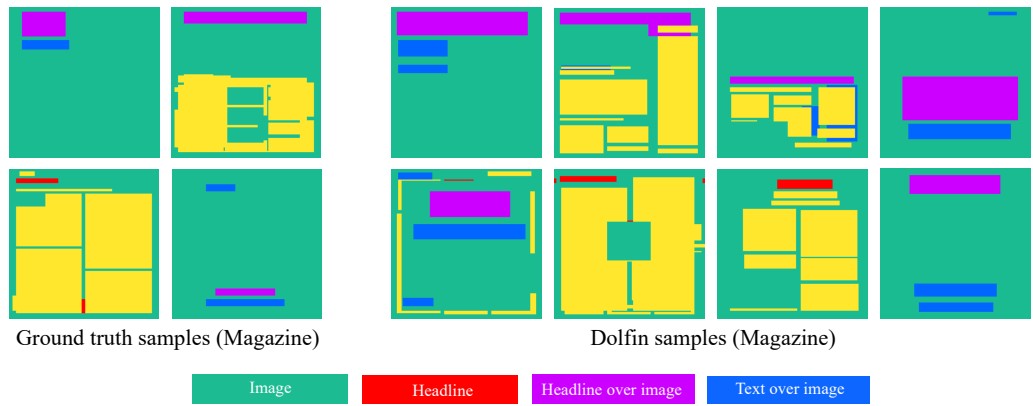

Ground truth samples (Magazine)      Dolfin samples (Magazine)

| Image | Headline | Headline over image | Text over image |

Figure 10: The left 4 figures are sampled from the Magazine dataset while the right 4 figures are generated results. We can see that our samples satisfy both quality and diversity.

## B.3 TEXTLOGO3K

TextLogo3K dataset contains about 3,500 text logo images. We adopt the official datasplit of the dataset. We present the numerical results for 4 metrics (MaxIoU, alignment, DocSim, FID) on the dataset in Table11. The visual results are shown in Figure11.

Table 11: TextLogo3K dataset quantitative results

| Method | MaxIoU ↑ | Align. ↓ | DocSim ↑ | FID ↑ |
|---|---|---|---|---|
| Dolfin | 0.76 | 1.29 | 0.14 | 12.43 |
| Dolfin-AR | 0.73 | 0.93 | 0.15 | 18.64 |

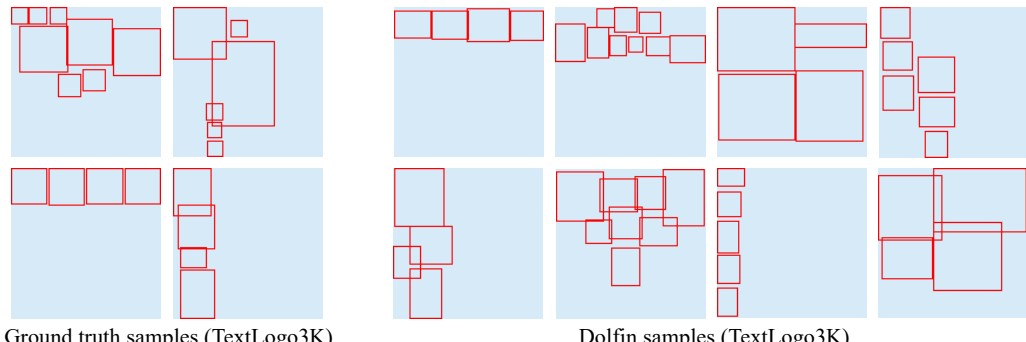

Ground truth samples (TextLogo3K)     Dolfin samples (TextLogo3K)

Figure 11: The left 4 figures are sampled from the TeskLogo3K dataset while the right 4 figures are generated results. We can see that our samples satisfy both quality and diversity.

## C   LINE SEGMENT GENERATION

### C.1   GENERATING IMAGES USING CONTROLNET BASED ON THE GENERATED LINE SEGMENTS

We utilize the line segments generated by Dolfin to synthesize images using pre-trained ControlNet model Zhang & Agrawala (2023). Results are shown in Figure 12 to illustrate both the quality and diversity achieved through this approach.

### C.2   DIFFERENCE: THE NEW METRIC TO EVALUATE LINE SEGMENT GENERATION

We introduce the *Difference* score as an additional metric to the FID score (Heusel et al., 2018) in order to evaluate the similarity between two sets of images consisting same number of line segments. (Each image can be considered a set of line segments.)

For a pair of normalized line segments $l_1 = ((x_{11}, y_{11}), (x_{12}, y_{12}))$ and $l_2 = ((x_{21}, y_{21}), (x_{22}, y_{22}))$, the inter-segment weight is computed as

$$w_{line}(l_1, l_2) = |x_{11} - x_{21}| + |y_{11} - y_{21}| + |x_{12} - x_{22}| + |y_{12} - y_{22}|.$$

For two images, $I_1$ and $I_2$, let $L_1$ and $L_2$ denote the sets of line segments in the two images, respectively. We employ the Hungarian Matching algorithm to find matches between $L_1$ and $L_2$ that satisfy the minimum sum of $w_{line}$ for each matching pair, with the inter-image weight $W_{image}(I_1, I_2)$ defined as the sum.

For the two sets of images, $S_1$ and $S_2$, we once again utilize the Hungarian Matching algorithm to find matches between them that satisfy the minimum sum of $W_{image}$ for each matching pair. The "Difference" score is defined as the average of the inter-image weights $W_{image}$.

### C.3   VISUAL RESULTS COMPARISON OF LINE SEGMENT GENERATION

We present the visual results of line segment generation by Dolfin model as well as LT (Gupta et al., 2021) and LayoutGAN++ (Kikuchi et al., 2021) in Figure 13. The result indicates that our model can outperform the competing methods.

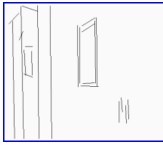

Sampled Line Segment

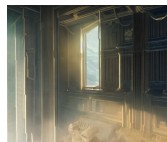

A picture of a room with big windows and columns, a detailed matte painting by Noah Bradley, cgsociety, retrofuturism, matte painting, concept art, dystopian, in the style of warhammer 4 0 k

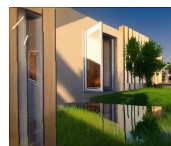

A picture of a house, outdoor, nature, tall tree, trees, shorter grass, shorter street, daytime, light effect, rippling reflections, warm lighting, vivid, beautiful, trending on artstation, 4k, unreal engine, high quality, highly detailed

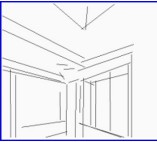

Sampled Line Segment

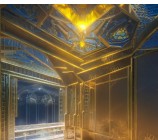

A picture of the inside of a crystal palace garden with gold intricate details, by Frank Lloyd Wright, concept art, inrincate, sharp focus, digital painting, unreal engine, cgsociety, neoclassical, mech, robot, fractal

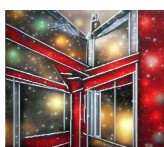

A picture of an outside wall decorated with Christmas themed objects is outside on the lawn, outside window is decorated with Christmas flowers, outside window is decorated with Christmas trees. outside on the

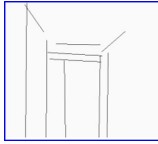

Sampled Line Segment

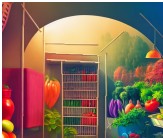

A picture of a garden with fruit on the shelf, shelves of vegetables, bright colors, cinematic, 4k, concept art, trending on artstation

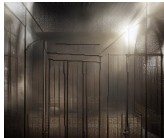

A picture of a cage, dark color, highly detailed, dark background, digital painting, artstation, concept art, illustration

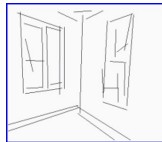

Sampled Line Segment

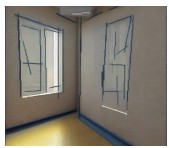

A picture of a classroom with 2 windows, realistic, 8k, highly detailed, trending on artstation, bokeh

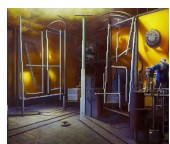

A picture of a chemistry lab full of grungy chemicals and machines, by George Caleb Bingham and Anato Finnstark, Norman Rockwell, Tim Hildebrandt, Tim White, Moebius, Bruce Pennington, donato giancola, larry elmore, oil on canvas, masterpiece, trending on artstation, featured on pixiv, cinematic composition, dramatic pose, beautiful lighting, sharp, details, hyper-detailed, HD

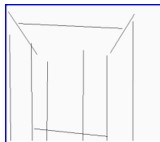

Sampled Line Segment

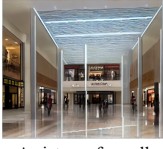

A picture of a mall

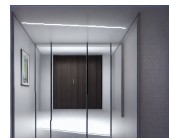

A picture of an office, realistic, 8k, extremely detailed, cgi, trending on artstation, hyper - realistic render, 4K,bella, full body, path traced, house background

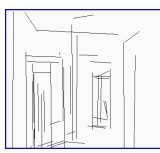

Sampled Line Segment

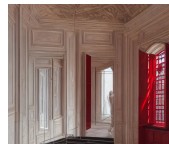

A picture of a museum with a bright red window, Trending on Artstation, featured on Behance, well-rendered, intricate, highly detailed, intricate, innocent, epic composition, concept art, sharp focus, colorful, 8K, art by WLOP and Artgerm and Greg Rutkowski and Alphonse Mucha

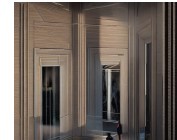

A picture of the outside of a tall building, realistic, 4k, breathtaking, beautiful composition, Ilya Kuvshinov, Greg Rutkowski, Zhang Jingna-H 640

Figure 12: The first image in each row represents a sample of line segments generated by our DOLFIN model. The two subsequent images in each row are generated by feeding the canny image from the first column into the ControlNet Zhang & Agrawala (2023) model, with the captions indicating the prompts used for generation.

Dolfin Samples     Ground Truth Samples     LT Samples     LayoutGAN++ Samples

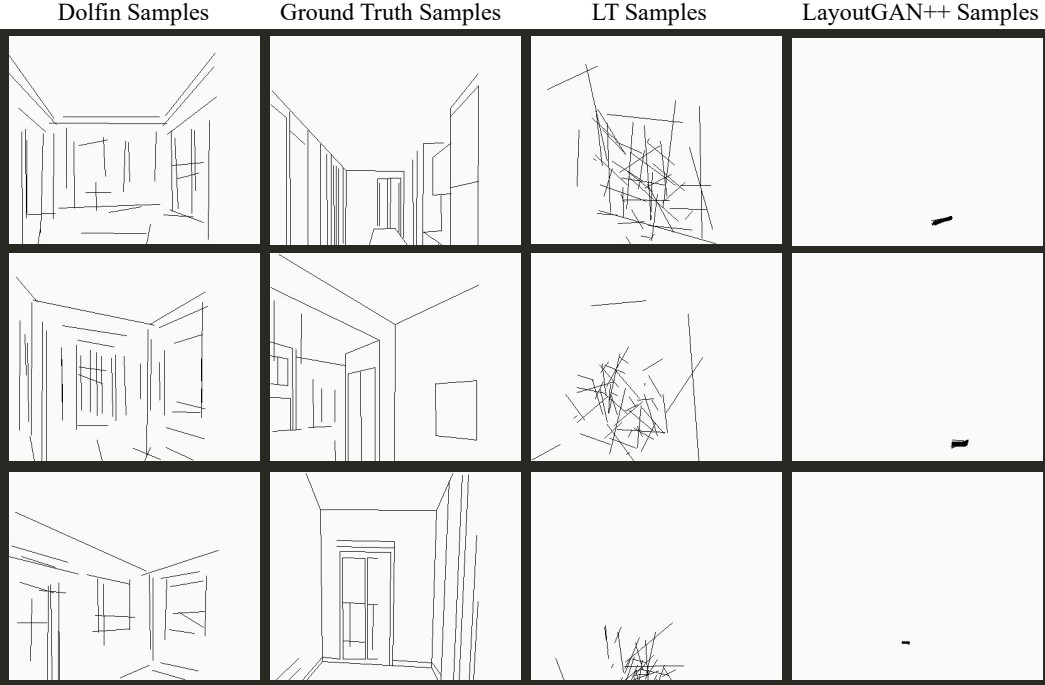

Figure 13: Each column contains 3 images from each set, including Dolfin samples (ours), ground truth samples, LayoutTransformer samples, and LayoutGAN++ samples. We can see that our samples achieve the highest quality among the samples from the 3 models. Since the competing methods are not designed for line segment generation, it is difficult for them to generate high quality line segments.

