# OpenReview forum: "Dolfin: Diffusion Layout Transformers without Autoencoder"
_ICLR.cc/2024/Conference — ICLR 2024 Conference Withdrawn Submission_

### Official Review · Reviewer_qsqM · 2023-10-28

**Soundness:** 2 fair
**Presentation:** 2 fair
**Contribution:** 2 fair
**Rating:** 3
**Confidence:** 4

**Summary:**

This paper introduces a diffusion-based layout generation model utilizing transformers. It removes the autoencoder layer typically used in a diffusion model for layout/image generation and directly operates on the layout input space. The proposed two model variants (Dolfin and Dolfin-AR) empirically improves performance across various metrics.

**Strengths:**

1. This paper is clearly written and easy to follow.
2. The proposed models notably improve quantitative results against generative layout benchmarks.

**Weaknesses:**

1. The main difference with previous models is by operating directly on the input space of layouts (the coordinates and corresponding class labels) instead of processing the layouts with VAE/dedicated modules. However the reasons for the brought-in performance gains are not sufficiently justified.
2. "enhancing transparency and interoperability" is overclaimed since it is a property of the standard diffusion process itself.
3. From the paper presentation it is not clear what are the modifications to the original DiT transformer other than omitting a category input.

**Questions:**

1. Why is operating on the original layout space better, especially when processing such data with dedicated neural modules is quite standard ? e.g. other than mentioned related works also standard in other generative models such as [1]. Could it be that the training data is insufficient?
2. Please check the metric arrow directions in Table 3, 4, 5.
3. In Fig.6, the generated samples exhibit some obvious unnaturalness (e.g. blue frames, bottom left, the window lines). Similar patterns exist in Fig.13. Is it because of insufficient training ? Could you compare it with PLay?

[1] GLIGEN: Open-Set Grounded Text-to-Image Generation

---

> ### Author Response · Authors · 2023-11-16
>
> Thank you.
>
> By directly operating on the input space, the model can enhance transparency, which is beneficial for performance. Additionally, failing to operate directly on the original space leads to unsatisfactory results in line segment generation. In diffusion-based layout generation, previous works have focused on latent space rather than the original space.
>
> **From the paper presentation it is not clear what are the modifications to the original DiT transformer other than omitting a category input.**
>
> We mainly do the following two modifications:
>
> (1) we omit the category input
>
> (2) we remove the VAE encoder and decoder that match the data to the original space.
>
> **Why is operating on the original layout space better, especially when processing such data with dedicated neural modules is quite standard ? e.g. other than mentioned related works also standard in other generative models such as [1]. Could it be that the training data is insufficient?**
>
> Please refer to the first paragraph that explains why operating on original space is better.In our paper, we focus on vectorized layout generation from scratch, using the same dataset as other works dealing with similar tasks. However, works like GLIGEN focus on generating images with text and additional conditions, a completely different task.
>
> **In Fig.6, the generated samples exhibit some obvious unnaturalness (e.g. blue frames, bottom left, the window lines). Similar patterns exist in Fig.13. Is it because of insufficient training ? Could you compare it with PLay?**
>
> The blue and red frames are added to make the white-background images clearer. The generated line segments generally follow the distribution of our training data (ShanghaiTech Wireframe dataset). In other works, such as PLay, they do not support direct operation on the input space, making it difficult to handle line segment tasks. What's more, they do not provide open-source code, which makes it difficult to reimplement their algorithm on new tasks.

---

### Official Review · Reviewer_SyW2 · 2023-10-28

**Soundness:** 3 good
**Presentation:** 3 good
**Contribution:** 2 fair
**Rating:** 5
**Confidence:** 4

**Summary:**

Diffusion Layout Transformers without Autoencoder (Dolfin) is proposed, with an efficient bi-directional (non-causal joint) sequence representation. An autoregressive diffusion model (Dolfin-AR) is also proposed to capture rich semantic correlations for the neighboring objects. The method is validated on 2D layout generation and line segment generation tasks.

**Strengths:**

- not requiring the autoencoder layer in the diffusion model
- autoregressive diffusion model to capture the rich semantic correlation between objects/items
- experiment on generating geometric structures beyond layout, such as line segments

**Weaknesses:**

- not using auto encoder is not a new idea, Imagen model is processing directly on pixels
- there is no intuition on why auot-regressive design leads to better semantic correlation, although this is observed from experiments
- not many baselines comparison for the line segment generation

**Questions:**

- explain the intuition of the advantage of auot-regressive design
- compare with image diffusion results for line generation, line representation can be obtained followed by a line detector
- each object in a layout is represented by a 4 × 4 tensor, why we need 4 entires for the entire layout width/height? Once it's normalized, is that always -1/1?
- in Algorithm 1 and 2, is it better to use a different index than "t" in the for loop? The for loop index has different meaning than the diffusion step t.

---

> ### Author Response · Authors · 2023-11-16
>
> Thank you.
>
> **Q1: not using auto encoder is not a new idea, Imagen model is processing directly on pixels**
>
> The original resolution in Imagen is the 1024x1024 images, yet they employ a super-resolution model to upscale from 64x64 to the original 1024x1024 resolution. Therefore, claiming they operate on the original resolution is inaccurate. Furthermore, our work focuses mainly on vectorized structured data, which differs from the pixel data in Imagen.
>
> **Q2: explain the intuition of the advantage of auot-regressive design**
>
> By implementing the autoregressive model, each token, corresponding to a bounding box, has a tighter association with previous tokens, enhancing the model's alignment capabilities.
>
> **Q3: compare with image diffusion results for line generation, line representation can be obtained followed by a line detector**
>
> We generate the line segments from scratch, whereas line detectors require an original image to generate them. To our knowledge, there is no existing work focused on this generation task.
>
> **Q4: each object in a layout is represented by a 4 × 4 tensor, why we need 4 entires for the entire layout width/height? Once it's normalized, is that always -1/1?**
>
> The layouts in the datasets have different heights and widths. There is no parameter set to normalize them. Instead, the actual heights and widths are divided by 1000.

---

### Official Review · Reviewer_eh4o · 2023-11-01

**Soundness:** 2 fair
**Presentation:** 2 fair
**Contribution:** 1 poor
**Rating:** 3
**Confidence:** 4

**Summary:**

The paper introduces "Dolfin," a generative model that uses a transformer-based diffusion process for layout generation. The proposed method directly applies on the input space of the geometric objects. The method benefits from bi-directional representation and consists of two versions, the non-auto regressive version that process all tokens simultaneously and the auto-regressive version that predicts each token sequentially. The authors provided experiments on RICO and PublayNet datasets for layout generation task as well as additional experiments on Line Segments Generation.

**Strengths:**

The paper is detailed and easy to follow.

Additional experiments on line segment generation can be useful to consider along with the other tasks.

**Weaknesses:**

The paper offers potential value to the community. However, concerns regarding its novelty and the robustness of its experimental evaluations need to be addressed for it to be ready for publication.

Novelty: The core proposition of the paper, which involves the utilization of the input coordinate space for layout design generation through continuous diffusion models, is not entirely novel. Similar approaches have been discussed in prior works such as [1, 2].

Experiments and Comparison: The experiments presented currently lack comprehensiveness. The results in Table 1 do not facilitate a fair comparison between the proposed method and existing methods. Although Tables 2 and 3 provide more data points, they restrict their focus to MaxIoU and Alignment scores. Furthermore, the results suggest that the proposed method underperforms compared to the baselines. This underscores the need for a more in-depth analysis and comparison.


[1] LEGO-Net: Learning Regular Rearrangements of Objects in Rooms, CVPR 2023
[2] HouseDiffusion: Vector Floorplan Generation via a Diffusion Model with Discrete and Continuous Denoising, CVPR 2023

**Questions:**

Considering the large batch sizes which are used in the experiments (10k and 6k) compared to the conventional batch sizes up to 2048, adding a table on the effect of the batch size on the final result can be very insightful.

I couldn't find any direct comparison on pros and cons of the Dolfin and Dolfin-AR. It is better if you also add both versions to other tables as well.

---

> ### Author Response · Authors · 2023-11-16
>
> Thank you. We hope our responses adequately address the concerns regarding the novelty of our work.
>
> Before the advent of diffusion models and the widespread use of transformers, structural data generation mainly relied on CNN-based models such as VAE and GAN. However, these methods often operated on latent spaces, potentially leading to the loss of the data's structural features. In early diffusion image model development, U-Net was commonly used as the denoising backbone instead of Transformers, requiring the mapping of data to a continuous latent space or the application of techniques to manage aspects of discrete data. Following the introduction of Transformer-based diffusion (DiT), a series of studies have been inspired by it. However, they still adhere to DiT's original design, mapping inputs into a continuous latent space. We propose a transformer-based diffusion model dedicated to general layout generation. This model operates directly on the space of geometric items, improving explainability and facilitating easier adaptation by other models.
>
> **Q1: The core proposition of the paper, which involves the utilization of the input coordinate space for layout design generation through continuous diffusion models, is not entirely novel. Similar approaches have been discussed in prior works such as [1, 2].**
>
> LEGO [1] mainly focuses on using additional constraints to rearrange the vectorized input data instead of generating vectorized data from scratch. HouseDiffusion [2] uses graph-based layouts instead of bounding boxes, diverging from our method. Besides, it deals with layouts in separate (continuous and discrete) sections, potentially impacting overall structural coherence.
>
> **Q2: Experiments and Comparison: The experiments presented currently lack comprehensiveness. The results in Table 1 do not facilitate a fair comparison between the proposed method and existing methods. Although Tables 2 and 3 provide more data points, they restrict their focus to MaxIoU and Alignment scores. Furthermore, the results suggest that the proposed method underperforms compared to the baselines. This underscores the need for a more in-depth analysis and comparison.**
>
> For a fair comparison, we use the data from papers that provide a clear methodology for computing metrics. Play and UniLayout are not open-sourced. The authors of Play responded to us with details. Because implementation details lead to significantly different results, for example, adopting Play's approach using 1024 samples to compute scores results in an FID score of 1.73 for conditional generation (conditioned on category), which is better than the number presented in the LayoutDM paper. However, using only 512 samples, the score becomes 4.21, which is worse than LayoutDM. While our model underperforms the baselines in some circumstances, our method generally performs better overall.
>
> **Q3: Considering the large batch sizes which are used in the experiments (10k and 6k) compared to the conventional batch sizes up to 2048, adding a table on the effect of the batch size on the final result can be very insightful.**
>
> We have tried to train the model using batch sizes of 10k, 5k, 3k, 1k. The outcomes indicate that all configurations yield similar FID scores within the range of ± 0.2, which can be attributed to the randomness of the diffusion sampling process.
>
> **Q4: I couldn't find any direct comparison on pros and cons of the Dolfin and Dolfin-AR. It is better if you also add both versions to other tables as well.**
>
> For unconditional generation, we provide results for Doldin and Dolfin-AR. Dolfin outperforms in metrics like the FID score, while Dolfin-AR excels in the alignment score. Regarding conditional generation, since Dolfin-AR does not support it, we only present results for Dolfin.